# Auto-Instruct: Automatic Instruction Generation and Ranking for Black-Box Language Models

**Zhihan Zhang**✉♠*, **Shuohang Wang**◇, **Wenhao Yu**♠, **Yichong Xu**◇, **Dan Iter**◇,
**Qingkai Zeng**♠, **Yang Liu**◇, **Chenguang Zhu**◇, **Meng Jiang**♠
♠University of Notre Dame
◇Microsoft Azure AI
zzhang23@nd.edu

## Abstract

Large language models (LLMs) can perform a wide range of tasks by following natural language instructions, without the necessity of task-specific fine-tuning. Unfortunately, the performance of LLMs is greatly influenced by the quality of these instructions, and manually writing effective instructions for each task is a laborious and subjective process. In this paper, we introduce Auto-Instruct, a novel method to automatically improve the quality of instructions provided to LLMs. Our method leverages the inherent generative ability of LLMs to produce diverse candidate instructions for a given task, and then ranks them using a scoring model trained on a variety of 575 existing NLP tasks. In experiments on 118 out-of-domain tasks, Auto-Instruct surpasses both human-written instructions and existing baselines of LLM-generated instructions. Furthermore, our method exhibits notable generalizability even with other LLMs that are not incorporated into its training process.[1]

## 1 Introduction

Instruction-tuned large language models (LLMs) have gained considerable popularity as solutions to a myriad of NLP tasks, owing to their proficiency in interpreting natural language instructions (Wei et al., 2021; Chung et al., 2022; Ouyang et al., 2022; Taori et al., 2023). As fine-tuning LLMs often becomes unfeasible, instructions play an increasingly crucial role in prompting such black-box LLMs. Especially in the *true few-shot* [2] setting (Perez et al., 2021) where the user aims to tackle a new task with only a basic task description and a few data

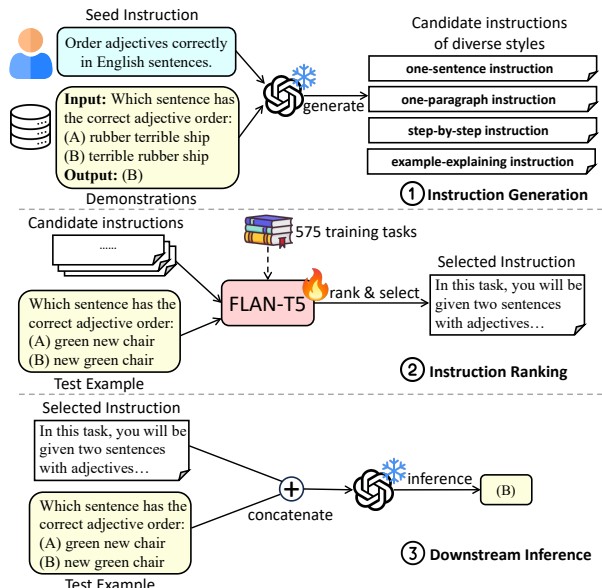

Figure 1: The Auto-Instruct pipeline. We first prompt the LLM to generate a diverse set of candidate instructions with different styles, and then train a model to rank and select the most effective instruction for a given example. Finally, the selected instruction is used to prompt the LLM to infer the output for this example.

examples at hand, a well-crafted instruction is imperative in enabling the LLM to grasp the required input-output mapping to complete the task.

Despite the significance of instructions, the prevailing approach when using a black-box LLM on a new task remains to be manual prompt engineering (White et al., 2023; Mishra et al., 2023). Such an approach, however, is not only time-consuming but also tends to yield suboptimal instructions. Against this backdrop, efforts have been made to empower LLMs to generate instructions automatically (Honovich et al., 2022; Zhou et al., 2022; Singh et al., 2022). These approaches feed the LLM a handful of examples and prompt it to generate an instruction based on these demonstrations. While such methods showcase the LLM's capability to generate coherent instructions (Honovich et al., 2022), only generating a single instruction

---

*This work was done when Zhihan was an intern at Microsoft Azure AI.

[1]Model and code are available at https://github.com/ytyz1307zzh/Auto-Instruct.

[2]A scenario where no additional training or validation data are available for hyperparameter tuning and prompt selection, in addition to the few-shot examples (Perez et al., 2021).

cannot guarantee reliable performance for unseen examples in the given task. As a straightforward solution, validation sets have been used to evaluate the effectiveness of a set of sampled instructions (Zhou et al., 2022; Singh et al., 2022), but this is impracticable for many tasks defined under the true few-shot setting (Suzgun et al., 2022). Besides, these approaches have primarily been tested on simple tasks where basic instructions are already sufficient, such as arithmetic operations or sentiment classification. More complex tasks in NLP benchmarks (Wang et al., 2022), which necessitate careful instruction engineering, remain largely unexamined for an automatic solution.

To address the aforementioned challenges, we propose Auto-Instruct, a novel approach to automatically generate and rank instructions for black-box LLMs across various NLP tasks, under the true few-shot setting. For each downstream task, we first prompt the LLM to sample a variety of candidate instructions, based on a basic seed instruction and few-shot demonstrations. We collect a diverse candidate set by specifying the expected *style* of each instruction. Recognizing the variable performance of LLMs across different instructions, coupled with the lack of validation data for pre-emptive instruction selection, we train a scoring model to rank and select the most appropriate instruction for each downstream test example. To ensure necessary generalizability in the few-shot setting, the model is trained on 575 exisiting NLP tasks before being deployed for out-of-domain test tasks. Finally, the selected instruction is used to prompt the LLM for downstream inference.

In experiments with OpenAI's *text-davinci-003*, Auto-Instruct yields remarkable performance on 118 out-of-domain tasks from Super Natural Instructions (SuperNI; Wang et al., 2022) and Big Bench Hard (BBH; Suzgun et al., 2022). Showing robust generalizability in out-of-domain scenarios, Auto-Instruct outperforms human-written seed instructions, the state-of-the-art instruction generation approach iPrompt (Singh et al., 2022), and various baselines of prompting the LLM for instruction selection. Moreover, Auto-Instruct exhibits impressive performance in the zero-shot setting and in generalization to other LLMs (*i.e.*, ChatGPT and GPT-4). Our study underlines that automatically generating and ranking instructions is a promising approach for leveraging the power of black-box LLMs effectively.

## 2 Related Work

The choice of instructions plays a pivotal role in effectively utilizing LLMs. To this end, a range of approaches has been implemented, with *parametric optimization* and *LLM-based generation* standing out as prominent methods. Parametric optimization primarily involves utilizing parameters to tune instructions (Shin et al., 2020; Shi et al., 2022; Deng et al., 2022). For instance, Shin et al. (2020) employed a gradient-based search over a predetermined length of discrete tokens as the instruction. Shi et al. (2022) further improved this approach by preserving the readability of the sampled tokens through a perplexity constraint. As a more flexible approach, Deng et al. (2022) optimized instruction generation through reinforcement learning, with rewards computed based on the LLM output. However, these strategies require access to either LLM parameters or a training set for optimization, making them less applicable to black-box LLMs with only a limited number of available examples. Moreover, instructions generated by these methods often lack fluency or even become gibberish, thereby compromising their interpretability.

In contrast, the LLM-based generation thread selects instructions by directly prompting the LLM (Honovich et al., 2022; Zhou et al., 2022; Singh et al., 2022). For example, Honovich et al. (2022) were among the first to reveal that LLMs could write an instruction for a given task after observing just a few demonstrations, and Zhou et al. (2022) improved the quality of the generated instructions by selecting the best performed one on the validation data. iPrompt (Singh et al., 2022) is the most capable method so far with its iterative generation and validation process for selecting instructions. Nevertheless, these approaches still necessitate a validation set for instruction ranking, and the instructions they generate typically underperform compared to those written by humans.

Besides the choice of instructions, researchers have also explored other orthogonal directions of improving LLM prompts, such as the selection of in-context demonstrations. Some works focused on identifying the most suitable demonstrations from training examples (Rubin et al., 2022; Lu et al., 2022a; Wang et al., 2023a) and their optimal ordering (Lu et al., 2022b) in the few-shot prompt. Other studies examined the engineering and selection of reasoning chains that are paired with the few-shot demonstrations on multi-step reasoning tasks (Wei

et al., 2022; Zhang et al., 2022b; Ye and Durrett, 2023; Liang et al., 2023b). We reserve the exploration of integrating these orthogonal techniques with our approach to holistically optimize the entire LLM prompt for future work.

# 3 Problem Formulation

In this work, we focus on the true few-shot setting where a user aims to tackle a new task with a black-box LLM. While it is easy to come up with a handful of examples and a basic description, the user may not have insights into what kind of instructions would be effective for unseen examples. Hence, given the few-shot examples as demonstrations and the basic description as a seed instruction, our goal is to automate the process of creating a more effective instruction for the given task.

We formulate our problem following the conventional practices of in-context learning (Dong et al., 2023). In the aforementioned **few-shot** setting, the *prompt* to query a black-box LLM comprises an *instruction* $I$, the test input $x$, and a few input-output pairs as *demonstrations* $\{x_i^d, y_i^d\}_{i=1}^n$. The LLM is expected to generate an output $y \sim P(\cdot|I, \{x_i^d, y_i^d\}_{i=1}^n, x)$. This work aims to automatically find a superior instruction $I'$ based on the human-written seed instruction $I^s$, thereby circumventing the need for substantial manual engineering. Besides, we also explore the **zero-shot** setting where no demonstrations are given to the LLM.

Despite the instruction potentially having multiple ways of integrating with the demonstrations and the test input, to reduce the complexity of the problem, we format the whole prompt in the order of $(I, x_1^d, y_1^d, \cdots, x_n^d, y_n^d, x)$. This aligns with the convention of problem-solving where the task is first outlined, followed by the provision of data examples, and the test input is finally provided. In practice, we use $n = 3$ for all tasks.

# 4 Auto-Instruct

Auto-Instruct is composed of two steps: instruction generation and instruction ranking. We first prompt the black-box LLM to generate a diverse set of candidate instructions (§4.1) for each downstream task. Next, we train a scoring model to rank all candidate instructions for each given test example, as different examples can benefit from different instructions (§4.2). Then, the top-ranked instruction is selected to prompt the black-box LLM on that specific test example for downstream inference.

Write a step-by-step instruction on how to solve the following task.

**Task**: [seed instruction]

**Examples**:
Input: [input of demonstration #1]
Output: [output of demonstration #1]
......
Input: [input of demonstration #n]
Output: [output of demonstration #n]

**Instruction**:

Figure 2: The meta-prompt that guides the LLM to generate a step-by-step instruction for the given task. Other meta-prompts are shown in Appendix E.

## 4.1 Instruction Generation

As mentioned in §3, we leverage a basic human-written task description as the seed instruction $I^s$ and prompt the black-box LLM to generate a number of candidate instructions $\{I_j^c\}_{j=1}^m$. Specifically, in the few-shot setting, we prompt the LLM to generate candidate instructions $I^c \sim P(\cdot|I^s, \{x_i^d, y_i^d\}_{i=1}^n)$ based on the seed instruction and few-shot demonstrations. Previous approaches (Zhou et al., 2022; Singh et al., 2022) only utilized a single meta-prompt[3] and collected candidate instructions via token sampling. Usually, such sampled instructions only show minor variations in phrasing rather than substantial content diversity. Moreover, their quality recursively rely on the arbitrary choice of the meta-prompt, which transfers the unreliability of manual instruction engineering to manual meta-prompt engineering.

In our improved approach, we curate a set of meta-prompts to stimulate the LLM to sample diverse candidate instructions by defining different required *styles* of the instruction. These meta-prompts include:

1. Write an instruction on how to solve the following task in *one sentence*.
2. Write an instruction on how to solve the following task in *one paragraph*.
3. Write a *step-by-step* instruction on how to solve the following task.
4. Write an instruction on how to solve the following task. The instruction must include the *explanations of the given examples*.

Alongside these 4 meta-prompts, we also bring in human-written instructions from existing NLP tasks as demonstrations to guide the generation of

---

[3]The prompt for the LLM to generate instructions.

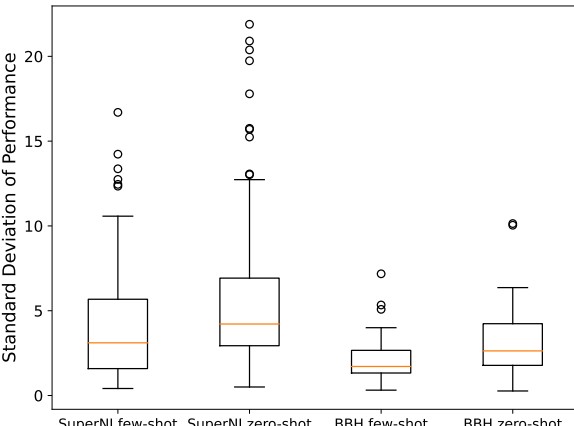

Figure 3: Box plot showing how much the LLM performance varies with different instructions, tested on OpenAI's *text-davinci-003*. Performance is evaluated by ROUGE-L on SuperNI and Accuracy on BBH. Each value represents the standard deviation of LLM performance across all generated instructions on a single task.

instructions. Intuitively, we prompt the LLM to emulate the style of human-written instructions in these demonstration tasks. We source demonstration tasks with their instructions from our training tasks in SuperNI, grouping them into 3 clusters based on the length of their instructions, so as to guide the LLM to generate instructions of different granularities. Figure 2 provides an example of the meta-prompt #3. Other meta-prompts are detailed in Appendix E.

Based on these 7 distinct meta-prompts (*i.e.*, 4 style-specific meta-prompts + 3 groups of demonstration tasks), we sample 3 instructions under each meta-prompt via nucleus sampling (Holtzman et al., 2020). Including the original seed instruction, we collect a total of 22 candidate instructions for each task. As a result, we create a diverse and comprehensive set of candidate instructions, thereby reducing the randomness brought by the nuances of different meta-prompts.

In the zero-shot setting, due to the absence of demonstrations, the LLM is prompted to generate the candidate instruction $I^c \sim P(\cdot|I^s)$ based solely on the seed instruction. Besides, the example-explaining meta-prompt is removed. As we demonstrate in §5.4.5, even without the aid of demonstrations, our style-specific meta-prompts still enable the LLM to generate informative instructions.

### 4.1.1 Instability Under Different Instructions

While LLMs are capable of generating meaningful instructions, relying on a single generated instruc-

tion will probably lead to suboptimal performance due to the LLM's sensitivity to the phrasing of the instructions. This instability is particularly evident in the zero-shot setting due to the lack of demonstrations to assist prediction. In Figure 3, we calculate the standard deviation of LLM performance using different instructions, after having evaluated all instructions for each downstream task. This indicates the expected performance fluctuation when substituting one instruction for another. The median standard deviation across all tasks are 3.1 and 4.2 points in ROUGE-L for few-shot and zero-shot settings respectively on SuperNI, and the upper quartiles are 5.7 and 6.9 points respectively. The choice of instruction even causes double-digit performance fluctuation on many tasks. Therefore, the development of a method to rank and select instructions becomes an essential undertaking.

### 4.2 Instruction Ranking

In a true few-shot setting, demonstrations are inadequate to reflect the effectiveness of candidate instructions due to the small sample size. To circumvent this limitation, we train a generalizable instruction ranking model across a variety of NLP tasks, and subsequently apply it to each test example in out-of-domain tasks. Intuitively, this model is trained to rank instructions against their downstream performance on the LLM, *i.e.*, to assign higher scores to more effective instructions.

#### 4.2.1 Model

Owing to the proven generalizibility of the T5 model family (Raffel et al., 2020; Sanh et al., 2022), we start from the instruction-tuned FLAN-T5-Large model (Chung et al., 2022) and train it with our instruction ranking objective. Given a specific example $(x, y)$ where $x$ is the input and $y$ is the ground-truth output, as well as an arbitrary candidate instruction $I^c$, the model predicts a score $Q_{T5}(I^c, x)$ as an estimate of the instruction's effectiveness on the example. Leveraging the instruction-following nature of FLAN-T5, we give the following prompt to the ranking model:

> Example: $x$. Input: $I^c$. Is this a good instruction to solve the example?

$Q_{T5}(I^c, x)$ is then calculated as the logit of the "yes" token at the starting position of the decoder. Additionally, we obtain the downstream performance of $I^c$ by calculating the ROUGE-L score between the LLM's predicted output $\hat{y}$ (when $I^c$ is

used as the instruction) against the groud-truth output $y$, denoted as $r(y, \hat{y})$. The model is then trained with a list-wise loss to align the scores $Q_{\text{T5}}(I^c, x)$ of all candidate instructions with their corresponding downstream performance $r(y, \hat{y})$, while considering relative superiority among different instructions. Specifically, we first normalize both the list of predicted scores $Q_{\text{T5}}(I^c, x)$ and the list of downstream performance $r(y, \hat{y})$ by applying softmax across all candidate instructions, and then compute the KL-divergence between these two normalized distributions as the training loss:

$$\mathcal{L} = \frac{1}{|\mathcal{B}|} \sum_{(x,y) \in \mathcal{B}} \mathbb{KL}\Big(\sigma\big(r\,(y, \hat{y})\big) \,\|\, \sigma\big(Q_{\text{T5}}\,(I^c, x)\big)\Big),$$
$$\text{where } \hat{y} \sim P_{\text{LLM}}(\cdot | I^c, \{x_i^d, y_i^d\}_{i=1}^n, x).$$

Note that $\mathcal{B}$ is a batch of examples and $\sigma$ is the softmax function. During testing, given a specific test example, among all candidate instructions, we select the $I^c$ that achieves the highest score $Q_{\text{T5}}(I^c, x)$ as the final instruction, and prompt LLM with it to obtain the desired output.

### 4.2.2 Training Data

To train such a ranking model with generalizability to out-of-domain tasks, we categorize the tasks in the SuperNI benchmark by their task type (*e.g.*, QA, sentiment analysis, etc.) and group these categories into training and test sets. We exclude tasks involving non-English languages or those with excessively long inputs. To avoid data leakage, we also exclude tasks from the training data which are sourced from the same dataset as any test task. This yields 575 tasks for training and 91 for testing. We sample up to 400 examples from each training task, which leads to 122k in total. Additional data pre-processing and filtering methods utilized to accelerate the training process can be found in Appendix A.

## 5 Experiments

### 5.1 Settings

To evaluate our approach under the true few-shot setting, we test it on a variety of out-of-domain tasks — 91 from SuperNI (Wang et al., 2022) and 27 from BBH (Suzgun et al., 2022), where there is no overlap between task categories in training and testing. The SuperNI test set comprises both classification and generation tasks, *e.g.*, commonsense classification, information extraction, etc[4]. BBH

presents a diverse set of tasks spanning commonsense QA and math problems. Average ROUGE-L[5] and exact-match accuracy are used for evaluation on SuperNI and BBH, respectively. Our main experiments are conducted using OpenAI's *text-davinci-003* for instruction generation and downstream inference. We also explored the instructions generated by ChatGPT (*gpt-3.5-turbo*) or GPT-4 (OpenAI, 2023) in §5.4.1.

In the zero-shot setting, the ranking model is separately trained on data where downstream ROUGE scores of candidate instructions are likewise obtained under zero-shot prompting. For zero-shot classification tasks, we append additional formatting instructions to the seed instruction to narrow down the answer options in both instruction generation and downstream inference. Additional experimental settings can be found in Appendix B.

### 5.2 Baselines

As baselines in our experiments, we first consider three alternative approaches based solely on prompting the LLM:

**(1) Cross-Validation**. We leverage the 3-shot demonstrations as validation data to rank the instructions, with each one acting as the test example iteratively while the other two serve as demonstrations. The ROUGE-L score (or accuracy for BBH) is used as the primary ranking criterion, and the log-probability of the ground-truth output is compared as tiebreaker. The instruction selected by the demonstrations is then applied on all test examples within the same task.

**(2) LM Selection**. We directly prompt the LLM itself to select an instruction by enumerating all candidate instructions in a single prompt. We number the instructions and ask the LLM to generate the number of the instruction it deems most suitable to each test example.

**(3) On-the-fly Generation**. As a simplified variant without instruction ranking, the model is asked to directly generate an instruction for each test example. The generated instruction is then used to prompt the LLM for the same example.

Furthermore, we consider **iPrompt** (Singh et al., 2022), the existing state-of-the-art approach in optimizing instructions with LLMs. iPrompt iteratively generates instructions until it cannot find one with better performance on a validation set. To evaluate

---

[4]The full list of SuperNI test tasks is in Appendix G.

[5]The original authors of SuperNI found ROUGE-L positively correlated to accuracy on classification tasks, so average ROUGE-L is applied for simplicity.

| Methods | Generation | Ranking | Few-shot | | Zero-shot | |
|---|---|---|---|---|---|---|
| | | | SuperNI | BBH | SuperNI | BBH |
| Empty Instruction* | None | None | 57.03 | 51.18 | 35.86 | 45.12 |
| Human Instruction* | Human | None | 60.94 | 50.30 | 46.81 | 45.59 |
| Random Selection† | LLM | Random | 61.61 | 50.88 | 45.80 | 45.98 |
| iPrompt* | LLM (*iterative*) | Examples | 57.08 | 50.46 | - | - |
| iPrompt+* | LLM (*iterative*) | Examples | 61.13 | 50.82 | - | - |
| Cross-Validation* | LLM | Examples | 62.02 | 51.20 | - | - |
| LM Selection† | LLM | LLM | 61.69 | 51.96 | 44.19 | 45.05 |
| On-the-fly Generation† | LLM | None | 61.03 | 51.38 | 45.85 | 45.47 |
| Auto-Instruct† | LLM | Trained Model | **64.35** | **52.04** | **49.50** | **47.35** |

Table 1: Results on SuperNI (91 tasks) and BBH (27 tasks) under the few-shot and zero-shot setting respectively. We report ROUGE-L on SuperNI and accuracy on BBH. Methods with * apply the same instruction for a certain task, while methods with † can select different instructions for different examples. iPrompt iteratively generates and ranks candidate instructions, while other methods adopt a generate-then-rank pipeline. We note that iPrompt, iPrompt+ and Cross-Validation are not applicable under the zero-shot setting due to the need of validation examples. Detailed results on SuperNI of different task categories can be found at Appendix D.1.

iPrompt under the true few-shot setting, we conduct its validation on the 3-shot demonstrations. Besides, since the original iPrompt generates instructions based on the examples without any task description, for a fair comparison, we implement an **iPrompt+** baseline that uses a similar meta-prompt to ours with the seed instruction (See Appendix C for details). In addition, we evaluate the performance of not using any instruction (**Empty Instruction**), directly using the human-written seed instruction (**Human Instruction**) or randomly selecting an instruction from the generated candidates (**Random Selection**) on each task.

### 5.3 Results

The overall results of SuperNI and BBH are shown in Table 1, where scores are averaged across all tasks. Auto-Instruct shows notable consistency and generalizability in out-of-domain scenarios, surpassing all baselines across different benchmarks and settings. Key findings are outlined below.

**The LLM shows competitive ability in generating effective instructions, yet ranking is still necessary.** In alignment with previous work (Zhou et al., 2022; Singh et al., 2022), the LLM is able to generate effective instructions for various tasks. Our style-specific meta-prompts enables the LLM to produce a diverse set of instructions to cater to varied scenarios where different tasks may favor different styles of instructions. In the few-shot setting, the LLM-generated instructions already

surpass their human-written counterparts on average, as indicated by the random selection scores. Although humans may have prior knowledge of some examples when they write the instructions, the LLM, not given any demonstrations in the zero-shot setting, generates instructions of comparable quality to those written by humans. Nevertheless, neither random selection nor directly generating a single instruction (*i.e.*, on-the-fly generation) significantly improves over the human-written baseline. This aligns with the instability of the LLM performance across different instructions as discussed in Figure 3, which indicates further instruction ranking is still essential.

**Simply prompting the LLM or using the validation data are not reliable in the low-resource setting.** Although offering the convenience of not training any models, both directly prompting the LLM (LM selection) and using few-shot demonstrations for validation (iPrompt and cross-validation) fail to deliver consistently improved results compared to random selection. This highlights that (1) the LLM itself lacks clue of the expected downstream performance of different instructions; (2) the volume of validation data must be substantial enough to effectively estimate the performance of instructions on the test data, which brings high cost in many realistic scenarios.

**Our trained instruction ranking model is the most effective approach to select instructions so far.** Although the data and instructions for out-of-

| Methods | ChatGPT | GPT-4 |
|---|---|---|
| *Few-shot, instructions from text-davinci-003* | | |
| Human | 60.39 | 67.31 |
| Random | 60.44 | 67.07 |
| Auto-Instruct | **62.88** | **69.45** |
| *Few-shot, instructions from* ChatGPT/GPT-4 | | |
| Human | 60.39 | 67.31 |
| Random | 60.44 | 66.77 |
| Auto-Instruct | **62.32** | **68.16** |
| *Zero-shot, instructions from* ChatGPT/GPT-4 | | |
| Human | 47.77 | 54.11 |
| Random | 46.22 | 53.06 |
| Auto-Instruct | **49.04** | **55.53** |

Table 2: SuperNI results of transferring Auto-Instruct to ChatGPT and GPT-4, using either (1) instructions generated by *text-davinci-003*, or (2) instructions generated by the same model as downstream inference (*i.e.*, ChatGPT or GPT-4). The instruction ranking model is still the one trained on *text-davinci-003* instructions.

| Methods | Selection Acc | | Win Rate | |
|---|---|---|---|---|
| | Top1 | Top5 | *vs.* Empty | *vs.* Human |
| Human | 45.25 | 70.35 | 22.43 | - |
| Random | 46.76 | 70.13 | 24.95 | 16.87 |
| Cross-Validation | 47.61 | 68.73 | 26.77 | 20.74 |
| LM Selection | 47.53 | 71.07 | 25.17 | 17.93 |
| Auto-Instruct | **52.54** | **73.10** | **29.51** | **23.89** |

Table 3: Evaluation of instruction ranking on silver labels. *Left*: we evaluate the percentage of cases where the selected instruction is the best (top-1) or is among top-5 candidates, according to the actual downstream performance. We note that there could be multiple instructions sharing the best score. *Right*: we check the percentage of selected instructions that outperform either the empty instruction or the human-written ones.

domain tasks are not seen by the ranking model, it exhibits promising generalizability in selecting effective instructions thanks to the training on hundreds of different tasks. For example, on the SuperNI benchmark, it outperforms random selection by 4% and 8% on few-shot and zero-shot settings respectively. Besides, our complete pipeline delivers a relative 6% improvement over the original human instructions in both few-shot and zero-shot settings, indicating that the human-written instructions still need improvement in many contexts.

## 5.4 Analysis

In this section, we delve deeper into the performance of our approach by analyzing the use of other LLMs for instruction generation, the performance on seen tasks, the size of training data, and case studies. Additional analysis of the comparison between Auto-Instruct and multi-answer ensemble is in Appendix D. These analyses are conducted in the few-shot setting unless stated otherwise.

### 5.4.1 Generalization to other LLMs

To further test the generalizability of our approach, we transfer Auto-Instruct to other LLMs by using ChatGPT (*gpt-3.5-turbo*) and GPT-4 as downstream inference models. As Table 2 suggests, instructions selected by Auto-Instruct on *text-davinci-003* are still effective if transferred to ChatGPT and

GPT-4. Furthermore, our instruction ranking model is able to rank instructions generated by ChatGPT or GPT-4 under both few-shot and zero-shot scenarios, despite not having seen any instruction created by these LLMs during training. Improved results can also be seen when transferring Auto-Instruct to LLaMA-2-chat (Touvron et al., 2023), a recent open-source LLM, as shown in Appendix D.2. As a conclusion, despite variations in phrasing across instructions generated by different LLMs, the underlying pattern determining instruction effectiveness is transferable, although the largest improvement is still seen in the same-LLM experiments. Suffice to say, our trained instruction ranking model can be directly applied to select instructions for other LLMs without the need of re-training.

### 5.4.2 Evaluation of Instruction Ranking

To investigate the effectiveness of the instruction ranking model, we compare it with other instruction selection baselines by assigning silver labels to candidate instructions, with results detailed in Table 3. First, we use the actual downstream performance of the candidate instructions as silver labels. Our ranking model is more capable of distinguishing better instructions, as shown by an evidently higher accuracy of picking the top-1 or top-5 instructions among all 22 candidates. Second, we evaluate how often the selected instruction improves the downstream performance in comparison to either the empty instruction or the human-written instruction. Once again, the instructions from our ranking model makes the most significant improvements, advancing the human-written counterparts in 7% more cases than random selection. The con-

| Methods | Unseen Tasks | Seen Tasks |
|---|---|---|
| Human | 54.59 | 40.32 |
| Random | 55.57 | 39.74 |
| Auto-Instruct | **60.18** | **45.89** |
| ⊢ (*vs.* Random) | **(+8.3%)** | **(+15.5%)** |

Table 4: Results on instruction-sensitive test data of both seen tasks (100 tasks seen in training) and unseen tasks (the same as Table 1) from SuperNI. We additionally report the relative improvement ratio to the random selection baseline since the vanilla performance is not on the same scale.

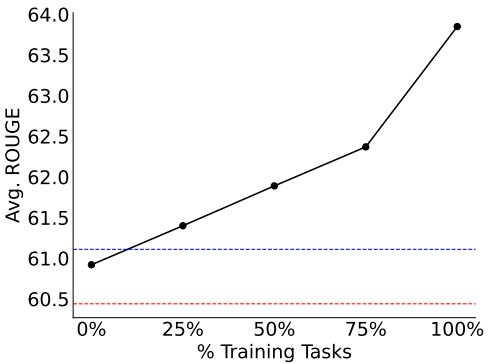

Figure 4: Results of using different number of training tasks. 0% means directly using the pre-trained FLAN-T5 checkpoint in instruction ranking, which shows a similar performance to random instruction selection.

sistent performance gain across all silver-label evaluations further corroborates the superiority of our model over alternative ranking methods based on cross-validation or LM selection.

### 5.4.3 Auto-Instruct on Seen Tasks

Besides the out-of-domain setting, we explore an in-domain setting where we select additional examples from tasks seen during training, so as to further examine the competency of the instruction ranking model. For a fair comparison of the model's ranking abilities across different tasks, we experiment with instruction-sensitive examples, defined as examples where not all candidate instructions yield the same ROUGE score. We sample 100 additional examples from each of 100 tasks that are seen in training but not included in the dev set. As presented in Table 4, the model shows enhanced ranking ability on seen tasks due to prior exposure to the instructions during training. This indicates that our approach is useful in both data-rich and data-scarce circumstances.

### 5.4.4 Effect of More Training Tasks

To analyze the effect of large-scale multi-task training on out-of-domain generalizability, we manipulate the number of training tasks of the instruction ranking model. Specifically, We exclude tasks from the training set by their category, *i.e.*, all tasks from selected categories are removed. As shown in Figure 4, the increment in the number of training tasks from additional categories is a key contributor to the superior performance of our model compared to the random selection baseline. Since the performance has not plateaued when all tasks are included, it is plausible to expect further performance gains if more training tasks are available.

### 5.4.5 Analysis of the Selected Instructions

Figure 6 illustrates how our selected instructions improve the original human instructions. As indicated by the average similarity scores, Auto-Instruct is able to provide instructions more similar to the optimal ones among the candidates. As demonstrated by the scatter plot, in scenarios where the selected instruction outperforms the human instruction, its embedding usually deviates significantly from that of the human instruction but stays close to the optimal one. These results suggest that the selected instruction refines the human-written seed instruction by progressing towards the ideal solution, while the embedding distance between the selected and seed instructions makes such improvement hard to achieve by pure manual engineering.

In addition, we offer a case study in Figure 5 in the zero-shot setting where the LLM cannot refer to any demonstrations. Nevertheless, the LLM manages to generate additional examples using the knowledge gained from its extensive pre-training. These additional examples can act as demonstrations to create a "2-shot inference" setting, leading to a correct prediction that could not be achieved via the original zero-shot inference. Conversely, we also present an example where the LLM-generated instruction includes hallucinated descriptions that distort the original meaning of the seed instruction. The mismatch between this instruction and the test example results in its rejection by the ranking model. Readers may find further case studies in Appendix F.

---

[6] www.sbert.net/docs/pretrained_models.html

**Case Study (Zero-Shot)**

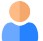 **Human Instruction:** Given a tuple, determine whether the Head is used for the Tail or not. Only answer with "Yes" or "No".

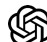 **Generated Instruction (Selected by the Ranking Model):** In this task, you are given a tuple which is a set of two elements, Head and Tail. Your task is to determine whether the Head is used for the Tail or not. To do this, you need to consider the meaning of the Head and Tail and determine whether the Head can be used to achieve the purpose of the Tail. For example, if the Head is "hammer" and the Tail is "nailing a nail", then the answer is "Yes" because a hammer can be used to nail a nail. If the Head is "hammer" and the Tail is "cutting a board", then the answer is "No" because a hammer cannot be used to cut a board. Only answer with "Yes" or "No".

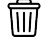 **Generated Instruction (Discarded by the Ranking Model):** To determine whether the Head is used for the Tail or not, you need to compare the two elements of the tuple. If the Head is the same as the Tail, then the answer is "Yes". If the Head is not the same as the Tail, then the answer is "No".

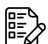 **Test Input:** Head: dental floss. Tail: provide dental hygiene.  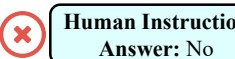 **Human Instruction Answer:** No  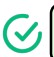 **Auto-Instruct Answer:** Yes

Figure 5: In this case, Auto-Instruct selects an instruction which "transforms" the zero-shot inference to a "2-shot" inference by providing additional examples (highlight in red), while discarding an instruction that includes hallucination in the task description (highlight in blue). The human instruction is also included in ranking candidates.

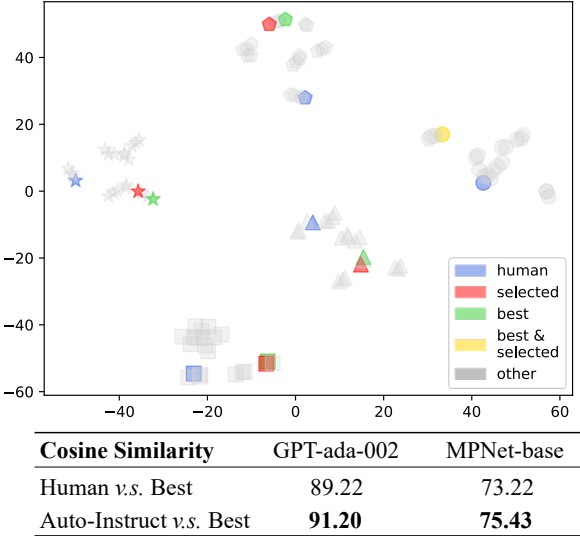

| Cosine Similarity | GPT-ada-002 | MPNet-base |
|---|---|---|
| Human *v.s.* Best | 89.22 | 73.22 |
| Auto-Instruct *v.s.* Best | **91.20** | **75.43** |

Figure 6: Above: Instruction embeddings of 5 SuperNI tasks where Auto-Instruct selected instruction performs better than human instruction, as visualized by T-SNE. "Best" refers to the instruction with the highest ROUGE score. Below: Average cosine similarity between instruction embeddings on all SuperNI tasks. Two embedding models are *text-embedding-ada-002* from OpenAI and *all-mpnet-base-v2* from Sentence-Transformers[6]. Best viewed in color.

## 6 Conclusion

In this work, we introduce Auto-Instruct, an automatic approach of generating, ranking and selecting instructions, which offers a solution to the high cost and subjectivity associated with human-engineered instructions. Our approach begins by prompting the LLM to generate a diverse set of candidate instructions. Next, an instruction ranking model trained on hundreds of tasks is used to rank the candidate instructions and select the most effective one to solve a specific example. Experimental results demonstrate that our approach provides better instructions than both human-written ones and those produced by previous instruction generation approaches, as tested on 118 out-of-domain tasks.

## Limitations

To our knowledge, this work has the following limitations:

- Due to the considerable cost associated with OpenAI models, and the limited capacity of their API interface, we only score the candidate instructions on a moderate number of tasks as described in §4.2.2. Given the results in Figure 4, we expect that the model could demonstrate improved generalizability if more training data with labeled instructions were available.

- The scope of this study is limited to the generation of instructions in English; tasks in non-English languages are not part of our training data. As a result, the model might not perform satisfactorily for non-English tasks. Further investigation into generating cross-lingual instructions is left for future work.

- Despite employing a wide range of meta-prompts, which significantly mitigates the dependence on prompt engineering, the phrasing of these meta-prompts could still influence the quality of the instructions generated. We leave the exploration of automatically diversify the generated instructions as future work.

## Acknowledgements

This work was supported by NSF IIS-2119531, IIS-2137396, IIS-2142827, IIS-2234058, CCF-1901059, and ONR N00014-22-1-2507. We thank Canwen Xu (University of California San Diego) for his valuable suggestions during paper writing.

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

## A  Training Data Pre-Processing

As detailed in §4.2, the instruction ranking model is trained to rank candidate instructions against their downstream performance on the LLM. The downstream performance of an instruction $I^c$ refers to how well the LLM's predicted output $\hat{y}$ matches the ground-truth output $y$ when using $I^c$ to prompt the LLM, as quantified by the ROUGE-L score $r(y, \hat{y})$. To calculate this score, we pair each training example with all 22 candidate instructions of the corresponding task (generated with the method in §4.1), and collect the LLM's predicted output to the example prompted by each candidate instruction. After calculating the ROUGE-L scores against the ground-truth, we discard examples where candidate instructions are not distinctly rankable – in cases where the range of downstream performance across different instructions is less than 10 points in ROUGE-L.

To accelerate the training process, we sample 8 candidate instructions from the total pool of 22 for each example, and train the model to rank these 8 instructions. However, in some tasks, certain instructions may significantly outperform others. Uniformly sampling 8 candidate instructions could result in such "extraordinary" instructions being disproportionately favored too many times in the training of the ranking model. To address this, we inversely proportion the sampling rate of each instruction to its popularity (*i.e.*, the number of cases where this instruction is superior to all others). Finally, we sample up to 400 examples from each training task, which leads to 122k training examples in total.

## B  Detailed Experimental Settings

The instruction ranking model is initialized with FLAN-T5-Large (780M parameters; Chung et al., 2022), and is trained using Adafactor (Shazeer and Stern, 2018) with learning rate 5e-5, batch size 128 and dropout rate 0.1. We employ an in-domain dev set including a total of 5k unseen examples from 100 training tasks to select the best checkpoint within 5 epochs. The validation performance on the dev set is 67.66 in ROUGE-L, while random selection only achieves a score of 54.28. When using OpenAI models, for instruction generation, we set the maximum instruction length to be 300 tokens, and we use a temperature of 1.0 and top_p of 0.75 for token sampling; for downstream inference, we set both to 0 for deterministic outputs. Gen-

```
Data:
Input: [input of demonstration #1]
Output: [output of demonstration #1]
......
Input: [input of demonstration #n]
Output: [output of demonstration #n]

Instruction:
```

Figure 7: The meta-prompt of instruction generation with iPrompt[7].

```
Write an instruction on how to solve the
following task.

Task: [seed instruction]

Examples:
Input: [input of demonstration #1]
Output: [output of demonstration #1]
......
Input: [input of demonstration #n]
Output: [output of demonstration #n]

Instruction:
```

Figure 8: The meta-prompt of instruction generation with iPrompt+, similar to ours in Figure 10.

erating all candidate instructions for 91 SuperNI test tasks cost us 18 USD in total, according to OpenAI's pricing (0.02 USD per 1k tokens for *text-davinci-003*). In *text-davinci-003* experiments, the random selection score is calculated as the average score across all instructions on each example, including the human-written seed instruction. In ChatGPT and GPT-4 instructions, due to the limited capacity of their API interfaces, we randomly sample an instruction for each example and test its performance.

## C   The iPrompt Baseline

In this section, we outline the adaptations made to the iPrompt[8] (Singh et al., 2022) method for our setting. We mainly address two discrepancies between its original implementation and our setup: (1) the original iPrompt generates instructions using GPT-J (Wang and Komatsuzaki, 2021), and (2) it uses a validation set to score and select instructions. To address (1), we use *text-davinci-003* for its instruction generation, identical to the model

used for downstream inference. For (2), we conduct its instruction validation on the 3-shot demonstrations. Due to the cost of iteratively requesting the OpenAI API, we incorporate an early stopping criterion which halts the process if the validation performance[9] has not improved for 10 iterations. Actually, almost all tasks stopped before 30 iterations. Following this, We select the instruction with the best validation performance to evaluate on the test examples.

According to the original codebase, we use the meta-prompt shown in Figure 7 for instruction generation with iPrompt. Since this meta-prompt does not utilize any task description, for a fair comparison, we implement an iPrompt+ baseline with a similar meta-prompt to our method which utilizes the seed instruction, as shown in Figure 8. Readers can refer to the original paper (Singh et al., 2022) for technical details of iPrompt.

## D   Additional Experimental Results

In this section, we present more experimental results in addition to those analyzed in Section 5. All experiments in this section are conducted in the few-shot setting unless stated otherwise.

### D.1   SuperNI Results by Task Category

Here, we present the detailed experimental results on 8 different categories of SuperNI test tasks (see Appedix G for the list of test tasks). As shown in Figure 9, Auto-Instruct surpasses the human-written and random instructions no matter it is evaluated on classification, extraction or generation tasks, with the only exception as answerability classification. Notably, Auto-Instruct outperforms the original human-written instruction by 10%, 9% and 8% on commonsense classification (classification tasks), word analogy (short generation tasks) and dialogue generation (long generation tasks), respectively.

### D.2   Generalization to Other LLMs

In addition to Section 5.4.1, we further assess the generalizability of Auto-Instruct to open-source LLMs. As demonstrated in Table 5, instructions selected by Auto-Instruct enhance the performance of LLaMA-2-chat (Touvron et al., 2023). This once again underscores the capability of Auto-Instruct

---

[7]In the original iPrompt implementation, the meta-prompt ends with the suffix `Prompt:`. However, this leads to incoherent instruction generation on our benchmarks. Therefore, we changed it to `Instruction:` which addressed this issue.

[8]`www.github.com/csinva/imodelsX/tree/master/imodelsx/iprompt`

[9]The average ROUGE-L score between the LLM's predicted output and the ground-truth on validation data.

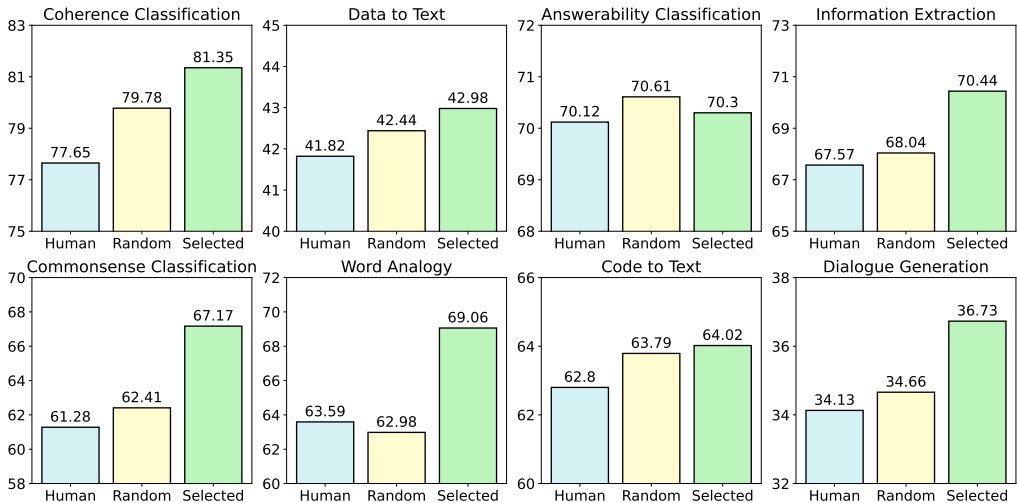

Figure 9: Few-shot performance of instructions selected by Auto-Instruct (denoted as "Selected") on all 8 categories of SuperNI test tasks, compared to human-written and random selected instructions.

| Methods | LLaMA-2-chat-7B |
|---|---|
| *Few-shot, instructions from text-davinci-003* | |
| Human | 53.87 |
| Random | 54.18 |
| Auto-Instruct | **55.90** |

Table 5: SuperNI results of transferring Auto-Instruct to LLaMA-2-chat-7B, using instructions generated by *text-davinci-003*. The instruction ranking model is still the one trained on *text-davinci-003* instructions.

| Method | Score |
|---|---|
| Human | 60.94 |
| Human (Ensemble) | 61.08 |
| Auto-Instruct | **64.35** |

Table 6: Results of multi-answer ensemble prompted by human-written instructions on SuperNI test tasks.

to generalize across different LLMs without retraining the instruction ranking model. It is worth noting that we use instructions generated by *text-davinci-003* in these experiments, because both the 7B and 13B versions of LLaMA-2-chat exhibit weaker abilities in following our meta-prompts for instruction generation, contrasted with mega-size GPT models. We leave the study of instruction generation with recent open-source LLMs as future work.

### D.3 Compare to Answer Ensemble

Given that Auto-Instruct includes sampling multiple candidate instructions before selecting the best one, we compare it to another sampling approach, *i.e.*, sampling and ensembling multiple answers. Using the original human-written instruction, we sample responses 10 times with nucleus sampling (Holtzman et al., 2020), without sampling multiple instructions. Then, we ensemble all 10 responses by marginalizing the LM probability of each unique response before selecting the most probable one, similar to the idea of self-

consistency (Wang et al., 2023b). The results, shown in Table 6, indicate that the answer ensemble approach only brings a marginal improvement on SuperNI, which is not comparable to the performance gain achieved with Auto-Instruct.

### E Meta-Prompts for Instruction Generation

In this section, we list all meta-prompts utilized during instruction generation, as outlined in §4.1. For the zero-shot setting, we omit the "Examples" field in the meta-prompt to let the LLM rephrase the seed instruction. Besides, the meta-prompt with explanations to the demonstrations is not applicable in the zero-shot setting. The meta-prompt that uses other tasks as demonstrations (Figure 10e) is integrated with three groups of demonstration tasks, each varying in the average instruction length. Therefore, the LLM is prompted to generate instructions of similar granularity to the demonstration tasks. Demonstration tasks are sampled from SuperNI. In SuperNI, each task is paired with a concise *task summary* and a detailed *task definition* which is usually much longer. For each demonstration task, we use the

| Write an instruction on how to solve the following task in one sentence.

**Task**: [seed instruction]

**Examples**:
Input: [input of demonstration #1]
Output: [output of demonstration #1]
......
Input: [input of demonstration #n]
Output: [output of demonstration #n]

**Instruction**: | Write an instruction on how to solve the following task in one paragraph.

**Task**: [seed instruction]

**Examples**:
Input: [input of demonstration #1]
Output: [output of demonstration #1]
......
Input: [input of demonstration #n]
Output: [output of demonstration #n]

**Instruction**: | Write a step-by-step instruction on how to solve the following task.

**Task**: [seed instruction]

**Examples**:
Input: [input of demonstration #1]
Output: [output of demonstration #1]
......
Input: [input of demonstration #n]
Output: [output of demonstration #n]

**Instruction**: |
|---|---|---|
| (a) One-sentence instruction | (b) One-paragraph instruction | (c) Step-by-step instruction |

| Write a step-by-step instruction on how to solve the following task.

**Task**: [seed instruction]

**Examples**:
Input: [input of demonstration #1]
Output: [output of demonstration #1]
......
Input: [input of demonstration #n]
Output: [output of demonstration #n]

**Instruction**: | Write an instruction on how to solve the following task.

**Task**: [seed instruction of task #1]

**Examples**: [input-output demonstrations of task #1]

**Instruction**: [instruction of task #1]
......
**Task**: [seed instruction of task #N]

**Examples**: [input-output demonstrations of task #N]

**Instruction**: [instruction of task #N]
**Task**: [seed instruction of the test task]

**Examples**: [input-output demonstrations of the test task]

**Instruction**: |
|---|---|
| (d) Example explanation instruction | (e) Use other tasks as demonstrations |

Figure 10: Meta-prompts that we use to specify different desired styles of instructions during instruction generation. For Figure 10e, we collect 3 groups of demonstration tasks with varying average instruction length (short, medium, long), so as to guide the LLM to generate instructions of different granularities.

task summary as the seed instruction and the task definition as the target instruction. We abstain from utilizing the task definition in test tasks because (1) some task definitions are too long to fit in the T5 model together with the input (2) we practically find that the LLM tends to repeat the task definition to a large extent if it is used as the seed instruction. Although Auto-Instruct has never seen the much longer task definition of test tasks, our selected instruction still performs better than using the task definition as the instruction, which holds an average score of 62.41 on SuperNI in the few-shot setting. We leave the exploration of integrating more complicated instructions as future work.

## F   Additional Case Study

In this section, we provide 3 more cases (2 few-shot and 1 zero-shot) where Auto-Instruct improves the original human-written instructions. These case studies are shown in Figure 11, 12, and 13. Please refer to the corresponding captions for detailed case explanations.

## G   All Test Tasks

In Table 7, we list all 91 SuperNI test tasks used in our out-of-domain experiments. Since the size of tasks is imbalanced on SuperNI, for efficient evaluation, we randomly sample 200 instances for each task, making a total of 18,200 test examples.

**Case Study (Few-Shot)**

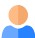

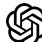
**Human Instruction:** Given a sentence from a medical study paper, select the tokens representing information about participants.

**Auto-Instruct:** Scan the sentence and look for information related to participants, *such as descriptions of age, numbers, or health conditions, and select the tokens that encapsulate this information.* For example, given the sentence "Participants were 58 children (46 boys) with autism between 3 and 4 years of age", the output should be "58 children (46 boys) with autism between 3 and 4 years of age."

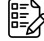
**Test Input:** To consider the impact on primary care patient outcomes of using both a screener to determine elevated anxiety levels among patients with previously undetected anxiety and a physician intervention to inform physicians of their patients' conditions .

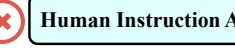
**Human Instruction Answer:** patients

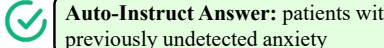
**Auto-Instruct Answer:** patients with previously undetected anxiety

Figure 11: A case study in the few-shot setting, where the few-shot demonstrations are omitted for brevity. The initial human-written instruction provides general guidelines for extracting participant information from a medical study. However, it does not specify the scope of such information, leading to an answer that only includes the keyword "patients" but ignores other relevant information. In contrast, Auto-Instruct provides an instruction that delineates the types of participant information to be extracted (highlight in red), after seeing the output formats of the demonstrations. Prompted by the improved instruction which suggests health conditions are examples of the requested information, the LLM generates a more comprehensive answer that incorporates the patients' health conditions, *i.e.*, "with undetected anxiety".

**Case Study (Few-Shot)**

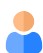
**Human Instruction:** Task description: Answer questions about which times certain events could have occurred.

**Auto-Instruct:** To solve this task, you must answer questions about when certain events could have occurred. To do this, you must analyze the information given in the question and the given examples.

For each example, you must analyze the information given about the events that happened before and after the event you are trying to find the time for. The question will usually give you the time that the person woke up and the time when the event they are going to closed. From this, you must determine the time frame in which the event could have occurred.

For example, in the first example, we know that Emily woke up at 1pm and the museum closed after 7pm. Therefore, the time frame in which Emily could have gone to the museum is between 1pm and 7pm. From the other information given, we know that Emily was at the library from 2pm to 4pm, at the theater from 4pm to 5pm, at the airport from 5pm to 6pm, and at the mall from 6pm to 7pm. Therefore, the only time frame in which Emily could have gone to the museum is between 1pm and 2pm. Therefore, the correct answer is (A)… *(explanations of other two demonstrations are omitted)*

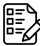
**Test Input:** Today, Jason went to the movies. Between what times could they have gone? We know that: Jason woke up at 10am. Linda saw Jason getting a coffee at the cafe from 10am to 3pm. Jennifer saw Jason walking towards the Statue of Liberty from 6pm to 7pm. Sean saw Jason buying a phone at the electronics store from 7pm to 10pm. The movies was closed after 10pm. Between what times could Jason have gone to the movies?
**Options:** (A) 3pm to 6pm (B) 10am to 3pm (C) 7pm to 10pm (D) 6pm to 7pm

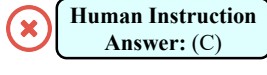
**Human Instruction Answer:** (C)

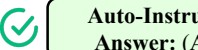
**Auto-Instruct Answer:** (A)

Figure 12: A case study in the few-shot setting, where the few-shot demonstrations are omitted for brevity. The human instruction provides a general and concise description of the question's requirements. In contrast, generated based on the demonstrations, the instruction from Auto-Instruct offers a more concrete description about the information present in the input and emphasizes which aspects should be focused on (highlight in red). Besides, Auto-Instruct provides explanations of the few-shot demonstrations as complementary information for the LLM to understand these examples.

**Case Study (Zero-Shot)**

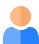 **Human Instruction:** Categorize the knowledge required to answer a pronoun coreference resolution question. Only choose your answer from "Quantity", "Object", "Others", "Property", "Spatial", and "Temporal".

**Auto-Instruct:** In this task, you will be given a pronoun coreference resolution question and you need to categorize the knowledge required to answer the question. You can only choose your answer from the provided list: "Quantity", "Object", "Others", "Property", "Spatial", and "Temporal". "Quantity" knowledge refers to the knowledge of quantities, such as how many, how much, etc. "Object" knowledge refers to the knowledge of the properties of objects, such as size, color, etc. "Others" knowledge refers to the knowledge of entities other than objects, such as people, places, events, etc. "Property" knowledge refers to the knowledge of the properties of entities, such as occupation, location, etc. "Spatial" knowledge refers to the knowledge of the spatial relations between entities, such as distance, direction, etc. "Temporal" knowledge refers to the knowledge of the temporal relations between entities, such as order, sequence, etc.

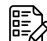 **Test Input: Sentence**: The painting in Mark's living room shows an oak tree. It is to the right of the bookcase. The "It" refers to the painting.
**Question**: What is the knowledge type needed to solve this relation?

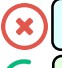 **Human Instruction Answer:** Spatial

**Auto-Instruct Answer:** Property

Figure 13: In this zero-shot classification case, the human-written instruction only provides the name of each category. As a result, the LLM can only attempt to determine the target category based on these single-word surface names, which often lack sufficient clarity for differentiation. In contrast, the instruction provided by Auto-Instruct explains the meaning of each category, which greatly facilitates the LLM's comprehension of these categories. While Auto-Instruct tends to over-interpret when explaining the "Others" category, most of the additional information (highlight in red) are useful for making more accurate predictions.

| Task Category | Task Names |
|---|---|
| Coherence Classification | task066_timetravel_binary_consistency_classification     task070_abductivenli_incorrect_classification 
 task1573_samsum_classification     task065_timetravel_consistent_sentence_classification 
 task298_storycloze_correct_end_classification |
| Data to Text | task1728_web_nlg_data_to_text     task1407_dart_question_generation 
 task677_ollie_sentence_answer_generation     task1409_dart_text_generation 
 task1598_nyc_long_text_generation     task957_e2e_nlg_text_generation_generate |
| Answerability Classification | task349_squad2.0_answerable_unanswerable_question_classification     task226_english_language_answer_relevance_classification 
 task020_mctaco_span_based_question     task290_tellmewhy_question_answerability 
 task1439_doqa_cooking_isanswerable     task1442_doqa_movies_isanswerable 
 task242_tweetqa_classification     task1624_disfl_qa_question_yesno_classification 
 task520_aquamuse_answer_given_in_passage     task050_multirc_answerability |
| Information Extraction | task1506_celebrity_minimal_dob_span     task1517_limit_classfication 
 task456_matres_intention_classification     task388_torque_token_classification 
 task1518_limit_answer_generation     task1410_dart_relationship_extraction 
 task676_ollie_relationship_answer_generation     task180_intervention_extraction 
 task749_glucose_reverse_cause_emotion_detection     task684_online_privacy_policy_text_information_type_generation 
 task958_e2e_nlg_text_generation_parse     task1413_dart_object_identification 
 task292_storycommonsense_character_text_generation     task578_curiosity_dialogs_answer_generation 
 task1597_nyc_slot_filling     task747_glucose_cause_emotion_detection 
 task678_ollie_actual_relationship_answer_generation     task1510_evalution_relation_extraction 
 task1451_drug_dose_extraction     task683_online_privacy_policy_text_purpose_answer_generation 
 task179_participant_extraction     task1411_dart_subject_identification 
 task181_outcome_extraction     task748_glucose_reverse_cause_event_detection 
 task621_ohsumed_yes_no_numerical_answer_generation     task647_answer_generation |
| Commonsense Classification | task1210_atomic_classification_madeupof     task1215_atomic_classification_capableof 
 task1216_atomic_classification_causes     task1202_atomic_classification_xneed 
 task136_winowhy_knowledge_categorization     task1196_atomic_classification_oeffect 
 task291_semeval_2020_task4_commonsense_validation     task1208_atomic_classification_xreason 
 task1206_atomic_classification_isbefore     task1197_atomic_classification_oreact 
 task1213_atomic_classification_desires     task116_com2sense_commonsense_reasoning 
 task1201_atomic_classification_xintent     task1198_atomic_classification_owant 
 task1212_atomic_classification_hasproperty     task1203_atomic_classification_xreact 
 task1214_atomic_classification_xwant     task1200_atomic_classification_xeffect 
 task1209_atomic_classification_objectuse     task1204_atomic_classification_hinderedby 
 task1207_atomic_classification_atlocation     task1205_atomic_classification_isafter 
 task1199_atomic_classification_xattr |
| Word Analogy | task1156_bard_analogical_reasoning_tools     task1159_bard_analogical_reasoning_containers 
 task1155_bard_analogical_reasoning_trash_or_treasure     task1157_bard_analogical_reasoning_rooms_for_containers 
 task1154_bard_analogical_reasoning_travel     task1158_bard_analogical_reasoning_manipulating_items 
 task1152_bard_analogical_reasoning_causation     task1153_bard_analogical_reasoning_affordance |
| Code to Text | task131_scan_long_text_generation_action_command_long     task129_scan_long_text_generation_action_command_short 
 task110_logic2text_sentence_generation |
| Dialogue Generation | task1603_smcalflow_sentence_generation     task1714_convai3_sentence_generation 
 task360_spolin_yesand_response_generation     task574_air_dialogue_sentence_generation 
 task565_circa_answer_generation     task576_curiosity_dialogs_answer_generation 
 task1600_smcalflow_sentence_generation     task1729_personachat_generate_next 
 task1730_personachat_choose_next     task361_spolin_yesand_prompt_response_classification |

Table 7: All SuperNI test tasks, grouped into different categories. These task categories are not seen during the training of the instruction ranking model. Besides, any task that is sourced from the same original dataset as any test task is excluded from training.