# OpenReview forum: "Auto-Instruct: Automatic Instruction Generation and Ranking for Black-Box Language Models"
_EMNLP/2023/Conference — EMNLP 2023 Findings_

### Official Review · Reviewer_MRyL · 2023-08-04

**Soundness:** 4

**Excitement:**

4: Strong: This paper deepens the understanding of some phenomenon or lowers the barriers to an existing research direction.

**Paper Topic And Main Contributions:**

This author proposes a solution to one of the prevalent problems in Large Language Models - what is the best instruction for a particular task? Unlike existing approaches, the authors make the task harder and also more realistic, given a task we have few input-output demonstrations (no validation data) and a possible human-written instruction.

The solution is simple yet effective.
1) The authors prompt the LLMs to generate diverse instructions for a given task (human seed problem description and optionally a few input/output demonstrations) by varying meta-prompts.
2) Later, they train an instruction ranking model on a subset of SuperNI dataset. The goal of the ranker is to predict the expected performance of a given instruction on the test example provided.
3) Given a test example, they select the instruction which would yield the best performance as predicted by the model

the results show better performance compared to the baselines.

**Reasons To Accept:**

The solution is simple yet effective. the comparison with baselines and existing approaches are complete. The approach demonstrates strong performance. The paper is well-written

**Reasons To Reject:**

The approach requires an instruction ranker model.

**Reproducibility:**

4: Could mostly reproduce the results, but there may be some variation because of sample variance or minor variations in their interpretation of the protocol or method.

**Reviewer Confidence:**

3: Pretty sure, but there's a chance I missed something. Although I have a good feel for this area in general, I did not carefully check the paper's details, e.g., the math, experimental design, or novelty.

---

> ### Author Rebuttal · Authors · 2023-08-29
>
> Thanks for the comments! We appreciate the acknowledgement of our contributions and the effectiveness of our method. We focus on a **true few-shot** setting where few-shot demonstrations are the only available examples besides the test set, without additional validation sets. According to results in Table 1, in the true few-shot setting, traditional ranking methods like validation with few-shot examples or using LLM itself are not reliable, while training a generalizable model is the best solution to rank and select instructions. Although a ranking model was trained, results on transferring Auto-Instruct to ChatGPT and GPT-4 (Table 2) showed that once the model was trained, it can be applied to select instructions on other models. Therefore, people can use our model when they have a tight budget.

---

### Official Review · Reviewer_5KfX · 2023-08-05

**Soundness:** 3

**Excitement:**

3: Ambivalent: It has merits (e.g., it reports state-of-the-art results, the idea is nice), but there are key weaknesses (e.g., it describes incremental work), and it can significantly benefit from another round of revision. However, I won't object to accepting it if my co-reviewers champion it.

**Paper Topic And Main Contributions:**

This paper improved the prompt-based generation of large language models in few- and zero-shot settings. This paper proposed Auto-Instruct, which comprised two techniques. First, the instruction itself is generated by a large language model. Compared to previous studies, Auto-Instruct generates multiple instructions by sampling a prompt from four prompt candidates that specify a style of instruction. Second, Auto-Instruct introduces a reranker to select the best prompt from the generated prompts. The experimental results on Super Natural Instruction and BigBench showed the effectiveness of Auto-Instruct.

**Reasons To Accept:**

- Few- and zero-shot text generation with instruction has attracted much attention because of publications of large language models.
- This paper carefully designed the training and test splits by dividing them in terms of task types to evaluate the capability of large language models in few- and zero-shot settings, which is known as the true few-shot setting.
- Auto-Instruct improved the performance on the famous benchmarks.

**Reasons To Reject:**

- The novelty of Auto-Instruct is the multiple prompts generation and their reranking because many previous studies proposed the instruction generation with large language models. However, the experiments did not investigate the correctness of the reranker:
  - The authors can report the results of an oracle reranker, which selects the best prompt after evaluating the results. Because this is an upper bound of Auto Instruct, the authors can discuss the effectiveness of the reranker with the upper bound.
  - The authors can give silver labels to prompts in accordance with whether a prompt was the best prompt or not, and whether a prompt improved the score in the current task. The performance of the proposed reranker model can be evaluated in the classification task of the silver labels.
- The improvement seems marginal. Especially on BigBench, introducing a reranker increased by 0.46 points (51.38 vs 52.04) in the few-shot setting and  1.88 points (45.47 vs 47.35) in the zero-shot setting on accuracy. The authors can report the results of statistical tests because there are many tasks and they can sample few-shot demonstrations multiple times in the few-shot setting.

**Reproducibility:**

3: Could reproduce the results with some difficulty. The settings of parameters are underspecified or subjectively determined; the training/evaluation data are not widely available.

**Reviewer Confidence:**

4: Quite sure. I tried to check the important points carefully. It's unlikely, though conceivable, that I missed something that should affect my ratings.

---

> ### Author Rebuttal · Authors · 2023-08-29
>
> Thanks for the constructive comments and suggestions! We will add the corresponding experiments (as follows) to the final version of the paper.
>
> **1. Regarding evaluation of the ranking model**
>
> **Q1. The performance of an “oracle” ranker**
>
> We tested an “oracle” ranker which always selects the best instruction and a “worst” ranker which always selects the worst one. However, the “oracle” performance is dependent on the number of sampled instructions. More generated instructions lead to higher oracle performance, but may also bring more noise to make the ranking more challenging. Thus, the “oracle” performance cannot fully reflect whether the instruction ranking model is good enough.
>
> The results on SuperNI are as follows. “Ranking performance” is the relative ratio that a method achieves between “worst” and “oracle”.
>
> | Method     | Few-shot Score | Ranking Performance | Zero-shot Score | Ranking Performance |
> |-----------|:----------:|:----------:|:-----------:|:----------:|
> | Oracle | 80.60    | 100% |  68.80 |  100% |
> | Auto-Instruct | 64.35  | 61.2% | 49.50 | 59.6% |
> | Random   | 61.61    | 54.7% |  45.80 | 51.8% |
> | Human    | 60.94    | 53.1% | 46.81 | 53.9% |
> | Worst    | 38.67   | 0% |  21.06 | 0% |
>
> This shows Auto-Instruct is able to find better instructions compared to random selection or human instruction. It’s worth noting that randomly selecting a generated instruction is a *competitive* baseline since it performed comparably with human-written instructions. Nevertheless, the problem of instruction ranking still has much space for future work to explore, given the potential of the oracle scores.
>
> **Q2. Evaluation with silver labels**
>
> Thanks for the suggestion! We tested with 2 sets of silver labels (macro-average on all SuperNI tasks):
>
> (1) whether the selected instruction achieved the best or top-5 downstream performance among all candidate instructions. Note that there could be multiple instructions that share the best score.
>
> | Method     | Is best? | Is in top-5? |
> |-----------|:-----------:|:----------:|
> | Auto-Instruct | **52.54%**  | **73.10%** |
> | Random   | 46.26%    |  69.61% |
>
> (2) whether the selected instruction improved the downstream performance. We consider the *improvement* over (i) empty instruction (ii) human instruction.
>
> | Method     | Better than empty? | Better than human? |
> |-----------|:-----------:|:----------:|
> | Auto-Instruct | **35.85%**  | **23.89%** |
> | Random   | 32.58%    |  16.58% |
>
> Therefore, Auto-Instruct significantly outperformed random selection, with +6.3% in selecting the best instruction and +7.3% in improving over human instruction.
>
> **2. Regarding the significance of improvement on BBH**
>
> Thanks for the suggestions! First, we would like to mention that the tasks in BBH have a larger distribution shift with those SuperNI tasks used in model training. For example, many tasks in BBH are multiple-choice questions with a considerable number (5+) of options, but all tasks in SuperNI are answered in a generative fashion. Therefore, a relatively smaller improvement is expected on BBH, compared to SuperNI test tasks.
>
> Regarding statistical tests, BBH has a fixed set of 3 few-shot demonstrations for each task. Thus, we conducted paired t-tests between our model and baselines by permuting the order of these 3 demonstrations, as the order also affects LLM performance [1]. After testing all 6 permutations, the results are as follows (mean $\pm$ standard deviation):
>
> | Method     | Score | p-value |
> |-----------|:----------:|:----------:|
> | Auto-Instruct | **52.06 $\pm$ 0.42**  |  -  |
> | Random   | 50.52 $\pm$ 0.29  | 0.0005 |
> | On-the-fly Generation    | 51.05 $\pm$ 0.35  | 0.0076 |
>
> Therefore, Auto-Instruct outperformed both random selection and on-the-fly generation with significance level $\alpha=0.01$.
>
> Alternatively, we conducted statistical tests on the zero-shot setting. Since we cannot manipulate demonstrations in zero-shot, we evenly split the data into 5 subsets and evaluated  the performance on each. The results are:
>
> | Method     | Score | p-value |
> |-----------|:----------:|:----------:|
> | Auto-Instruct | **47.35 $\pm$ 0.92**  |  -  |
> | Random   | 45.98 $\pm$ 0.62  | 0.0150 |
> | On-the-fly Generation    | 45.48 $\pm$ 0.86  | 0.0363 |
>
> Therefore, Auto-Instruct outperformed both random selection and on-the-fly generation with significance level $\alpha=0.05$. The variance is bigger in zero-shot tests because the whole dataset was divided into smaller subsets.
>
> **References**
>
> [1] Lu et al. Fantastically Ordered Prompts and Where to Find Them: Overcoming Few-Shot Prompt Order Sensitivity.

---

### Official Review · Reviewer_n9EG · 2023-08-14

**Soundness:** 4

**Excitement:**

3: Ambivalent: It has merits (e.g., it reports state-of-the-art results, the idea is nice), but there are key weaknesses (e.g., it describes incremental work), and it can significantly benefit from another round of revision. However, I won't object to accepting it if my co-reviewers champion it.

**Paper Topic And Main Contributions:**

In this paper the authors propose Auto-Instruct, a pipeline approach to automatically generate instructions. First, they prompt a black-box LLM to generate a diverse set of candidate instructions for a pool of downstream tasks. Second, they train a scoring model to rank candidate instructions. Last, the top ranked instruction is used for few/zero-shot inference.

For the ranking of candidate instructions they fine-tune a T5 model to align the scores given by the ranking model with the downstream performance of the LLM when using each instruction.

Their results on SuperNI and BigBench, show that LLMs are generate instructions that are more useful than human generated ones and that ranking those boosts performance.

**Questions For The Authors:**

- In line 326 the equation shows softmax over Qt5, but previously you say that Qt5 is the logit of the "yes" token. Is this correct? The softmax should be performed on top of the whole output vocabulary, right?
-  It is not super clear to me how inference works. How many instructions do you sample at inference time?
- Depending on the answer to the previous questions, how much is the overhead at inference time if the instructions have to be sample from the model?
- Related to previous one too, how does the method compare to best of N sampling without instructions?

**Reasons To Accept:**

- The paper is well written and their proposed ranking method improves results over random selection.

**Reasons To Reject:**

- All the experiments are performed on top of closed LLMs.
- It is not super clear which is the inference strategy.

**Reproducibility:**

4: Could mostly reproduce the results, but there may be some variation because of sample variance or minor variations in their interpretation of the protocol or method.

**Reviewer Confidence:**

3: Pretty sure, but there's a chance I missed something. Although I have a good feel for this area in general, I did not carefully check the paper's details, e.g., the math, experimental design, or novelty.

---

> ### Author Rebuttal · Authors · 2023-08-29
>
> Thanks for the valuable comments and suggestions! We will add the corresponding clarifications and experiments (as follows) to the final version of the paper.
>
> **1. Regarding closed / open-source LLMs:**
>
> Using GPT models is to follow previous approaches in instruction generation [1, 2, 3] which all used GPT models. Besides, early open-source LLMs, such as OPT and GLM, do not perform as well as InstructGPT on generating instructions [2], which is another reason that we mainly used text-davinci-003 in our experiments.
>
> Given more instruction-tuned open-source LLMs coming out since March 2023, we additionally experimented with using instructions selected by our ranking model to prompt the Llama-2-chat-7B model. The results on few-shot SuperNI are as follows, showing that Auto-Instruct is also able to improve the performance on Llama.
>
> | Method     | Score |
> |-----------|:----------:|
> | Auto-Instruct | **55.90**    |
> | Random   | 54.18    |
> | Human    | 53.87    |
>
> **2. Regarding questions on the inference strategy:**
>
> **Q1. Question on softmax.**
>
> Sorry for the confusion. Yes, $Q_{T5}(I^c,x)$ is the logit of the "yes" token and we used it as the ranking score of the candidate instruction $I^c$. As we mentioned in **line 323**, during training, $Q_{T5}(I^c,x)$ was normalized across candidate instructions $I^c$, which means the softmax normalization was applied on all instructions rather than the tokens in the vocabulary. This is to align the ranking scores of the list of instructions with their relative downstream performance $r(y,\hat{y})$ which was also normalized across instructions. We will revise this part to make it clearer.
>
> **Q2. The number of sampled instructions.**
>
> As we mentioned in **line 254**, we sampled 22 candidate instructions (3 instructions generated for each of the 7 meta-prompts, plus the original human-written seed instruction) for each task.
>
> **Q3. The overhead of instruction generation at inference time.**
>
> It took 25 mins to generate all candidate instructions on 91 SuperNI test tasks by calling text-dainvic-003 API with sequential API requests. More time can be saved using parallel API requests. This included 767k input tokens for the model and 180k output tokens to generate. This cost 18 USD in total (0.02 USD per 1k tokens for text-davinci-003).
>
> **Q4. Comparison to best of N sampling**
>
> Thanks for the constructive suggestion! We additionally tried the setting where we used the original human-written instruction and sampled the answer 10 times, without sampling multiple instructions. Then, we ensembled all 10 answers by a majority voting, similar to [4]. The results are as follows. Sampling and ensembling multiple answers brought little improvement on SuperNI, which is not comparable to Auto-Instruct.
>
> | Method     | Score |
> |-----------|:----------:|
> | Auto-Instruct | **64.35**    |
> | Human (Ensemble)   | 61.08   |
> | Human    | 60.94   |
>
> **References**
>
> [1] Honovich et al. Instruction Induction: From Few Examples to Natural Language Task Descriptions.
>
> [2] Zhou et al. Large language models are human-level prompt engineers.
>
> [3] Singh et al. iPrompt: Explaining Data Patterns in Natural Language via Interpretable Autoprompting.
>
> [4] Wang et al. Self-consistency improves chain of thought reasoning in language models.

---

### Meta-Review · Area_Chair_n2J1 · 2023-09-18

**Recommendation:** 3

**Metareview:**

Addresses the issue of the need for prompt engineering LLMs and seeks to generate better instructions/prompts using LLMs for tasks automatically to improve downstream performance. Through prompting, an LLM (text-davinci-003) generates diverse instructions, followed by a FLAN-T5 reranking model to select the best ones. There were concerns about novelty, the significance of the improvement and doubts about the ranking performance.

An unusual KL-divergence loss is used to align ROUGE-L scores of outputs with the classifier likelihood which has some similarity with BRIO (Liu et al, 2022), but this choice is not compared with the simpler approach of a vanilla classifier. Furthermore, the LLM reranker baseline may significantly improve simply by using GPT-4 instead of text-davinci-003.

Reliance entirely on automatic metrics for evaluating LLM outputs, such as ROUGE-L, while good to have, do not give a full picture that only human evaluation would complete. Including a human evaluation would significantly improve the confidence in the evaluation.

---

### Decision · Program_Chairs · 2023-10-07

**Decision:**

Accept-Findings

**Comment:**

Addresses the issue of the need for prompt engineering LLMs and seeks to generate better instructions/prompts using LLMs for tasks automatically to improve downstream performance. Through prompting, an LLM (text-davinci-003) generates diverse instructions, followed by a FLAN-T5 reranking model to select the best ones. There were concerns about novelty, the significance of the improvement and doubts about the ranking performance.

An unusual KL-divergence loss is used to align ROUGE-L scores of outputs with the classifier likelihood which has some similarity with BRIO (Liu et al, 2022), but this choice is not compared with the simpler approach of a vanilla classifier. Furthermore, the LLM reranker baseline may significantly improve simply by using GPT-4 instead of text-davinci-003.

Reliance entirely on automatic metrics for evaluating LLM outputs, such as ROUGE-L, while good to have, do not give a full picture that only human evaluation would complete. Including a human evaluation would significantly improve the confidence in the evaluation.